# 3D-printed eye model: Simulation of intraocular pressure

**Hidenaga Kobashi**[1,2]*, **Masaaki Kobayashi**[3]

**1** Toneasy Inc., Tokyo, Japan, **2** Department of Ophthalmology, Keio University, School of Medicine, Tokyo, Japan, **3** Stratasys Japan Co., Ltd., Tokyo, Japan

* himon@hotmail.co.jp

**Data Availability Statement:** All relevant data are within the manuscript.

**Funding:** This article is based on results obtained from a project, JPNP0401005, commissioned by the New Energy and Industrial Technology Development Organization (NEDO). The funders had no role in study design, data collection and

## Abstract

### Purpose

To develop artificial eye models using 3D printing and to evaluate the correlation between different corneal thicknesses and intraocular pressures (IOPs).

### Methods

We designed 7 artificial eye models using a computer-aided design system and fabricated them using 3D printing. Corneal curvature and axial length were based on the Gullstrand eye model. Hydrogels were injected into the vitreous cavity, and seven different corneal thicknesses (200 to 800 μm) were prepared. In this proposed design, we also produced different corneal stiffnesses. A Tono-Pen AVIA tonometer was used by the same examiner to perform five consecutive IOP measurements in each eye model.

### Results

Different eye models were ideally created using 3D printing. IOP measurements were successfully performed in each eye model. The corneal thickness was significantly correlated with IOP ($R^2 = 0.927$; $P<0.001$).

### Conclusion

The 3D-printed eye model is useful for evaluating IOP measurements. This technique might be a promising alternative to the conventional porcine eye model.

## Introduction

Three-dimensional (3D) printing technology, also known as additive manufacturing technology, is a technology that manufactures building blocks layer by layer from combinations of materials by applying three-dimensional data. The primary process involves delivery of a series of thin three-dimensional computer-aided design (CAD) or computed tomography cross-sections by the software to the printer for rapid imaging from bottom to top [1]. Three-

analysis, decision to publish, or preparation of the manuscript.

**Competing interests:** The authors have declared that no competing interests exist.

dimensional printing is an all-inclusive term for a variety of methods that use digital data to produce 3D objects made of various materials (both synthetic and organic) [2, 3]. This technology can rapidly produce customized devices and prostheses at low costs via simple development and design processes and is capable of a single item at a time, each unique in shape and design. Such products may be as diverse as instruments that aid in early detection of common ocular conditions, diagnostic and therapeutic devices built specifically for individual patients, 3D-printed contact lenses and intraocular implants, and models that assist in surgery planning, improve patient and medical staff education. The role of 3D printing in the field of ophthalmology and eyecare is evolving, particularly after the introduction of high-resolution pico- to micrometer-scale 3D printers. Several applications, such as orbital implants, ocular prostheses, intraocular devices, ophthalmic models, surgical instruments, and surgical simulation devices, have been developed [4, 5]. To the best of our knowledge, to date, there have been no studies on 3D-printed eyeballs simulating different intraocular pressures (IOPs). A high IOP can induce glaucoma in models. Generally, the central corneal thickness (CCT) significantly affects IOP readings obtained by tonometry [6]. Previous studies have induced a higher IOP model by changing the height of a drip stand in porcine eyes [7, 8], but this technique is limited to a wet laboratory curriculum. To simplify the simulation of the elevated IOP model, we developed artificial eye models by changing the CCT based on a 3D printing approach. A good understanding of this technology will be beneficial to glaucoma research. The aim of this study was to evaluate the association between different CCTs and the IOP in 3D-printed eyeball models.

## Methods

### Design of the eyeball model

The 3D-printed eyeball model was generated by CAD software and a simulation environment represented by ANSYS Workbench 19.1. In the current study, the eyeball consisted of the cornea, sclera, and vitreous humor to highlight the change in corneal thickness. The optical model was based on Gullstrand's simplified schematic eye [3]. Fig 1a shows the structure and main components of the eye model, which is available as parameterizable CAD geometry. When we manufactured the eyeball, the "Material Jetting" style was used as an additive manufacturing. This process is as follows:

1. Jet liquid ultra violet (UV) curable resin onto the print tray.

2. Use a roller to level the UV curable resin.

3. Use a UV lamp to harden the UV curable resin.

4. Repeat processes 1–3 and build the models.

The material shore hardness is changed by the material jetting pattern. The output of the UV lamp and the rotational speed of the roller (600 rpm) were held constant during printing. This means that the material shore hardness is not changed by the output of the UV lamp and the speed of the roller.

To easily measure the IOP using a Tono-Pen AVIA tonometer (Reichert Inc., Depew, New York, USA), we created a cuboid eye holder. The aqueous and vitreous humor were simulated by the injection of hydrogels into the vitreous cavity. To induce different corneal stiffnesses, seven different CCTs were prepared for every 100-μm increase, and the curvatures of the cornea and sclera were simultaneously varied (Fig 1b). To simply control the change in the IOP, we changed only the CCT in the current study. Some parameters were fixed as the default

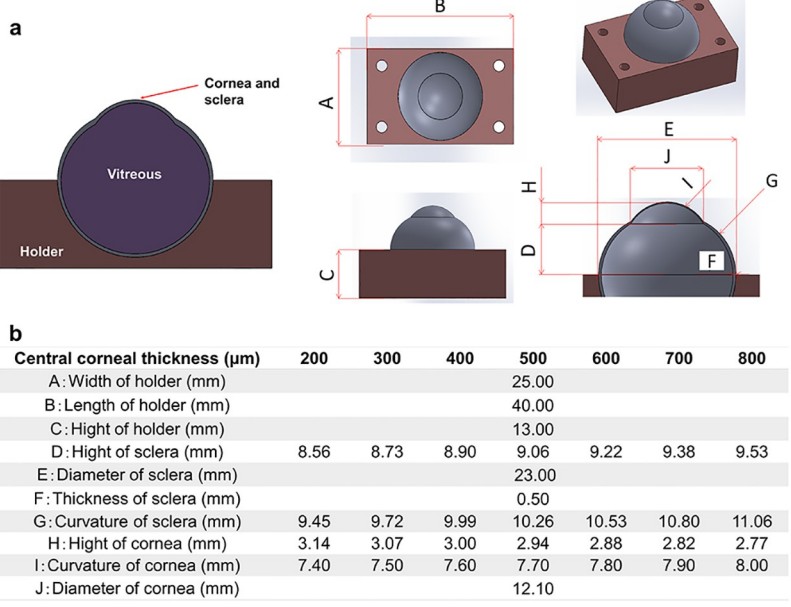

**Fig 1.** a) Schematic diagram of the 3D printed eye model as a computer-aided design construction. b) Characteristics of ocular biometric parameters in the cornea, sclera, and eyeball holder.

values. To assess the deviation between the printed specimen and the CAD design, we aimed to measure the actual value of the CCT using an ultrasound pachymeter (SP-100; Tomey, Nagoya, Japan). Three consecutive measurements were performed in each eye model.

In addition to the design of the eye geometry, an adequate description of the material properties is required for the 3D-printing simulation. Table 1 shows the material parameters of the eye model.

## Intraocular pressure measurements

Experiments with different CCTs in the eyeball model demonstrated different elastic responses when measuring pressure readings using the Tono-Pen AVIA tonometer. The range of IOP measurements was 5 to 55 mmHg in the tonometer. All series were taken five times by the same examiner (H.K.) using the tonometer at each corneal thickness level, and the mean value and standard deviation of each parameter were recorded. To determine the repeatability of the IOP measurement, intraclass correlation coefficients (ICCs) were calculated for the 5 repeated

**Table 1. Material parameters of the eye model.**

| | Cornea (500 µm) | Vitreous | Holder |
|---|---|---|---|
| Product name | Stratasys FLX9740-DM | Stratasys Gel Support | Stratasys VeroClear™ |
| Stiffness of material | Rubber-like | Gel | Hard |
| Tensile strength | 3–4 MPa | - | 50–65 MPa |
| Young's modulus of elasticity | Un open (industrial secrets) | - | 2000–3000 Mpa |
| Elongation break | 190–210% | - | 10–25% |
| Tensile tear resistance | 6.0–8.0 kg/cm | - | - |
| Flexural strength | Un open (industrial secrets) | - | 75–110 Mpa |
| Flexural module | Un open (industrial secrets) | - | 2200–3200 Mpa |

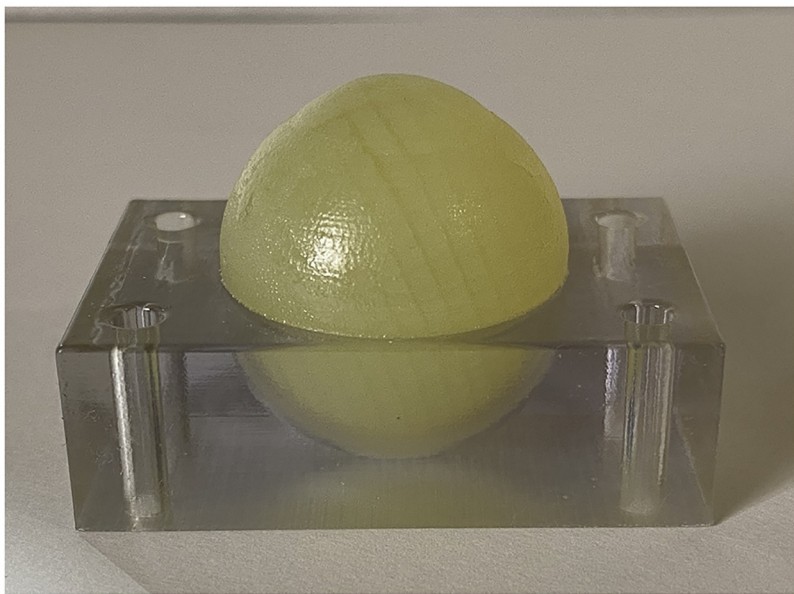

**Fig 2. Whole eyeball and holder simulated using 3D-printing technology.**

measurements at each corneal thickness. The ICC ranges from 0 to 1 and is commonly classified as follows: less than 0.75 indicates poor repeatability, 0.75 to less than 0.90 shows moderate repeatability, and greater than 0.90 represents high repeatability. The experiments were carried out at room temperature ($24 \pm 1$ °C).

## Data analysis

Analyses were performed with Statistical Analysis Software (version 9.4; SAS Institute, Cary, NC). Linear regression was employed to analyze the CCT in the model eye and IOP to evaluate the linear regression coefficient ($R^2$). The normality of all data was first checked by the Kolmogorov–Smirnov test. Because the use of parametric statistics was possible, the Pearson correlation coefficient was used to assess the correlation between the IOP and the CCT in each eye model. The outcome measures are reported as the mean ± standard deviation. A P value of $< 0.05$ was considered statistically significant.

## Results

Fig 2 shows a representative 3D-printed eyeball with a central corneal thickness of 500 μm. All eyeballs were successfully created at each corneal thickness. Table 2 presents the mean IOP of 3D-printed model eyeballs at each central corneal thickness. The repeatability of the IOP measurement was high, with an ICC of 0.969 (95% confidence interval: 0.909 to 0.994). The IOPs were significantly correlated with the CCTs ($R^2 = 0.927$, p<0.001) (Fig 3).

**Table 2. Intraocular pressure of the 3D-printed model eyeballs for each central corneal thickness.**

| Central corneal thickness (μm) | 200 | 300 | 400 | 500 | 600 | 700 | 800 |
|---|---|---|---|---|---|---|---|
| Mean (mmHg) | 9.6 | 18.0 | 22.4 | 23.6 | 23.8 | 30.6 | 35.2 |
| Standard deviation | 1.5 | 2.4 | 5.5 | 3.7 | 3.6 | 1.7 | 4.0 |

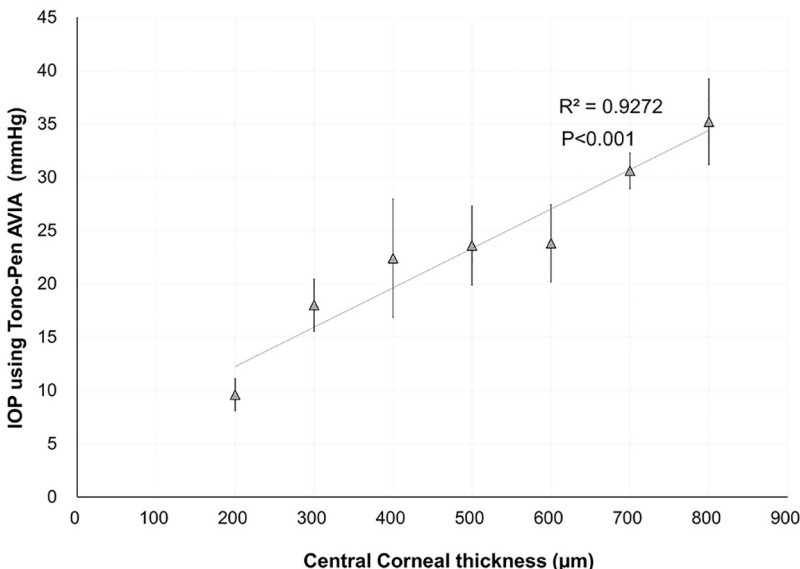

**Fig 3. Graph showing a significant correlation between central corneal thickness and intraocular pressure in the 3D-printed eyeball models.**

Table 3 presents the actual value of the CCT measured using an ultrasound pachymeter.

## Discussion

In this study, we successfully developed variable IOP eyeball models using 3D printing technology. In the 3D-printed eyeball model, the IOP increased as the CCT increased, which is similar to the results of a previous study in humans [6]. IOP measurements are dependent on the CCT. The current approach is the first to assess the correlation between IOP and CCT in 3D-printed eyeballs. This technique is quite unique and useful for basic research on ocular tonometers and glaucoma treatment because a special wet laboratory unit is needed. However, the true IOP might be unknown because it is impossible to determine the exact IOP that represents the pressure at the optic disc. Generally, corneal biomechanics play a role in the IOP in the human eye. Thus, we mimicked different IOP models with increasing CCTs using 3D-printed eye models. Almost all previous reports used ex vivo porcine eyeballs by changing the height of saline bottles to evaluate the elevated IOP [7, 8]. Our model might be a promising alternative to the conventional porcine eye model.

To simplify the methodology in this study, we changed the CCT and accordingly modified parameters such as the curvature and height of the sclera and cornea (Fig 1). We determined the diameter and thickness of the sclera at 23.00 mm and 0.50 mm, respectively. As a positive correlation was shown between the CCT and IOP in Fig 3, the impact of other geometric parameters might be smaller than that of corneal thickness.

**Table 3. Actual value of the central corneal thickness measured using ultrasound.**

| Central corneal thickness | | | | | | | |
|---|---|---|---|---|---|---|---|
| CAD design (μm) | 200 | 300 | 400 | 500 | 600 | 700 | 800 |
| Ultrasound pachymeter (μm) | 201.3 ± 7.6 | 303.0 ± 6.2 | 403.7 ± 7.2 | 504.7 ± 7.4 | 601.7 ± 12.1 | 699.7 ± 11.5 | 803.0 ± 2.0 |

With regard to the repeatability of the IOP measurements using the Tono-Pen AVIA, we confirmed the good repeatability for 5 consecutive measurements by the same examiner, as the standard deviation was almost within 5 mmHg at each corneal thickness. Our results of the reputability of the Tono-Pen AVIA in 3D-printed eyeballs are similar to those reported in the human eye [9]. However, it is difficult for the Tono-Pen AVIA to determine the correctness of the IOP measurements in 3D-printed eye models because the Tono-Pen AVIA itself has not been calibrated and designed to be used with those models. It would have been more beneficial to compare the IOP measured with other tannometers, such as iCare and noncontact tonometers. With regard to the reliability of the CCT in the CAD model, we evaluated the actual value of the CCT using an ultrasound pachymeter. The difference in the CCT between the CAD model and the ultrasound measurements was clinically acceptable because of the small difference (within 5 μm), as shown in Table 3.

There are three limitations to this study. First, the accurate morphological structure of the eyeball was not realized in the 3D-printed model because no crystalline lens, anterior chamber, or iris was created in this study. However, the primary aim was to develop a 3D-printed eyeball and simulate different IOP models with various corneal thicknesses. Second, precise corneal biomechanics were not achieved in these rubber-like model eyes. Based on the Gullstrand model of the cornea, we simulated stiffness, strength, and resistance values similar to those of an artificial cornea, as shown in Table 1. Further investigation is necessary to develop a 3D-printed cornea for IOP evaluation. Third, our 3D-printed eyeball models did not include iris, aqueous humor, or lens components, all of which affect IOP dynamics in humans, due to technological limitations.

In conclusion, a method for creating an elevated IOP model was demonstrated in a 3D-printed eyeball model. The model was useful for the assessment of the IOP in terms of basic research on glaucoma.

## Author Contributions

**Conceptualization:** Hidenaga Kobashi, Masaaki Kobayashi.

**Data curation:** Hidenaga Kobashi, Masaaki Kobayashi.

**Formal analysis:** Hidenaga Kobashi, Masaaki Kobayashi.

**Funding acquisition:** Hidenaga Kobashi.

**Investigation:** Hidenaga Kobashi.

**Methodology:** Hidenaga Kobashi, Masaaki Kobayashi.

**Software:** Hidenaga Kobashi, Masaaki Kobayashi.

**Validation:** Hidenaga Kobashi, Masaaki Kobayashi.

**Visualization:** Hidenaga Kobashi, Masaaki Kobayashi.

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
