## [Decision Letter · Decision Letter 0]

20 Oct 2022

PONE-D-22-236133D-printed eye model: simulation of intraocular pressurePLOS ONE

Dear Dr Kobashi,

Thank you for submitting your manuscript to PLOS ONE. After careful consideration, we feel that it has merit but does not fully meet PLOS ONE’s publication criteria as it currently stands. Therefore, we invite you to submit a revised version of the manuscript that addresses the points raised during the review process.

We look forward to receiving your revised manuscript.

Kind regards,

Aparna Rao

Academic Editor

PLOS ONE

Journal Requirements:

“This article is based on results obtained from a project, JPNP0401005, commissioned by the New Energy and Industrial Technology Development Organization (NEDO).”

“The author(s) have made the following disclosure(s): H.K.: Employee and equity owner, Toneasy Inc.; Patent, Toneasy Inc. No other disclosures were reported.”

Reviewers' comments:

Reviewer's Responses to Questions

**Comments to the Author**

1. Is the manuscript technically sound, and do the data support the conclusions?

Reviewer #1: No

Reviewer #2: Partly

2. Has the statistical analysis been performed appropriately and rigorously? 

Reviewer #1: N/A

Reviewer #2: Yes

3. Have the authors made all data underlying the findings in their manuscript fully available?

Reviewer #1: Yes

Reviewer #2: Yes

4. Is the manuscript presented in an intelligible fashion and written in standard English?

Reviewer #1: Yes

Reviewer #2: No

5. Review Comments to the Author

Reviewer #1: This research paper describes the development of a 3D eye model for testing the impact of varying corneal thickness on intraocular pressure. As is already known, corneal thickness has a significant influence on IOP readings obtained by almost any tonometer. Hence the finding that CCT influences IOP readings is not in any way novel. To my understanding, the other rationale for undertaking this study is the dependence on wet labs for studying the impact of various conditions on IOP measurements. However, the authors have not fully described the limitations of current methods and how their method addresses these limitations. This study should have included an in silico simulation assessing the impact of various elements of the setup (e.g., the mechanical holder, material properties geometrical attributes) before proceeding to the printing stage. My other major comments are as follows;

1. Although the main varying parameter is central corneal thickness, other parameters such as curvature of the sclera and cornea also change with each model. This is a significant limitations as not only CCT but also the geometry is likely to affect the IOP readings.

2. There is no description of the additive manufacturing (FDM?, SLA?) method that was used to manufacture the eyeball.

3. There is no mention of elastic modulus of the material that was used to print the cornea and the holder. Furthermore, as the cornea is a viscoelastic material, a rubber-like material (hence a purely elastic material) would not mimic corneal mechanical behavior.

4. The authors should have reported intraclass coefficient of serial measurements instead of describing repeated measurements as having "good repeatability".

5. Although acknowledged as a limitation, failure to factor in lens, iris, aqueous humor is a significant limitation hindering the applicability of this approach in studying IOP dynamics ex vivo.

Reviewer #2: The authors present a new methodology to study the impact of corneal thickness on the intraocular pressure (IOP) measurements. Overall, the paper addresses a quite interesting problem; on the other hand, there are several issues which must be addressed prior to the paper acceptance.

At first, the introduction and general contextualization is quite poor. For example, the authors cite that numerous published studies have considered 3D printing techniques in Health Sciences; however, just a few were cited. Besides, it would be interesting to mention other studies which considered using 3D printing to bring digital models to real life and then perform tests on them. They may consider, for example:

- Combining Microtomography, 3D Printing, and Numerical Simulations to Study Scale Effects on the Permeability of Porous Media by Luan C. de S. M. Ozelim and André L. B. Cavalcante

- A 3D additive manufacturing approach for the validation of a numerical wall-scale model of catalytic particulate filters by Igor Belot, Yixun Sun, David Vidal, Martin Votsmeier, Philippe Causse, François Trochu, François Bertrand.

English level should be enhanced. Sometimes, it becomes a bit confusing to fully understand the sentences.

Regarding the 3D printed model, more information should be provided about the elastic properties of the materials used. For example, which are the Young moduli of the materials? This, together with the corneal thickness, would probably have an impact on the IOP measurements. Besides, it is important to fully compare the mechanical properties of the printing materials to the ones of real human eyes.

The authors should provide more information about the printing procedure. Which were the steps? Also, they should present more details about the printer and its specs (minimum detail, minimum wall thickness etc).

A key aspect which needs to be at least discussed is how the authors assessed the quality of the printed specimens. In short, what are the expected deviations between the printed specimen and the one in the CAD design? Will these deviations impact on the effective corneal thickness? I do not know if the authors have access to a CT scan, but if they do, scanning a few of the printed samples would bring a valuable validation to the thicknesses considered in their analyses.

The authors mention the R-squared and the p-values obtained for the linear fit. Which hypothesis test is the p-value related to?

Since the authors have a complete control over the model, are they capable of measuring the “true” exact IOP?

Since Tono-Pen AVIA has not been calibrated and designed to be used with the 3D printed materials, this would be interesting to actually confirm the measurements are correct. Indicating that the results are reproducible is not sufficient to indicate the correctness of the result itself.

Finally, do the authors have control over the IOP change? How? Are the changes observed only due to the CCT changes? Any additional pressure is later applied to the gels? It is not clear in the paper how the IOP change occurs.

The issues above must be addressed to enhance the quality of the paper and make it publishable in such a high standard journal as PLOS One.

6. PLOS authors have the option to publish the peer review history of their article (what does this mean?). If published, this will include your full peer review and any attached files.

Reviewer #1: **Yes: **Eray Atalay

Reviewer #2: No

---

## [Author Response · Author response to Decision Letter 0]

9 Feb 2023

Dr. Aparna Rao,

Thank you for your e-mail regarding our manuscript (PONE-D-22-23613) titled “3D-printed eye model: simulation of intraocular pressure” as well as for the comments from the reviewer. We believe that the paper has been much improved, which was largely a result of the referees’ many thoughtful comments. We would like to respond below to each comment.

To Reviewer #1

This research paper describes the development of a 3D eye model for testing the impact of varying corneal thickness on intraocular pressure. As is already known, corneal thickness has a significant influence on IOP readings obtained by almost any tonometer. Hence the finding that CCT influences IOP readings is not in any way novel. To my understanding, the other rationale for undertaking this study is the dependence on wet labs for studying the impact of various conditions on IOP measurements. However, the authors have not fully described the limitations of current methods and how their method addresses these limitations. This study should have included an in silico simulation assessing the impact of various elements of the setup (e.g., the mechanical holder, material properties geometrical attributes) before proceeding to the printing stage. My other major comments are as follows;

Thank you for your comments on the revision. I have responded the following the comments as below.

1. Although the main varying parameter is central corneal thickness, other parameters such as curvature of the sclera and cornea also change with each model. This is a significant limitations as not only CCT but also the geometry is likely to affect the IOP readings.

To simplify the methodology in this study, we changed the CCT and accordingly modified the parameters such as curvature and height of sclera and cornea (Figure 1). We determined the diameter and thickens of sclera at 23.00 mm and 0.50 mm, respectively. As the positive correlation was shown between CCT and IOP in Figure 3, the impact of other geometric parameters might be smaller than that of corneal thickness. 

[Page 8, lines 143-147]: Three sentences have been added in Discussion.

“To simplify the methodology in this study, we changed the CCT and accordingly modified parameters such as the curvature and height of the sclera and cornea (Figure 1). We determined the diameter and thickness of the sclera at 23.00 mm and 0.50 mm, respectively. As a positive correlation was shown between the CCT and IOP in Figure 3, the impact of other geometric parameters might be smaller than that of corneal thickness.”

2. There is no description of the additive manufacturing (FDM?, SLA?) method that was used to manufacture the eyeball. 

This method of additive manufacturing is “Material Jetting”.

This process is the following:

1. Jetting some liquid materials which are ultra violet (UV) curable resin on the print tray.

2. Making level UV curable resin by roller.

3. Hardening UV curable resin by UV lamp.

4. Repeat process 1-3 and building models.

Material shore hardness is changed by material jetting pattern. Output of UV lamp and 600 rpm of roller are constant during printing. It means Material shore hardness isn’t changed by output of UV lamp and rpm of roller.

[Page 4, lines 71-80]: Four sentences have been added.

When we manufactured the eyeball, the “Material Jetting” style was used as an additive manufacturing. This process is as follows:

1. Jet liquid ultra violet (UV) curable resin onto the print tray.

2. Use a roller to level the UV curable resin.

3. Use a UV lamp to harden the UV curable resin.

4. Repeat processes 1-3 and build the models.

The material shore hardness is changed by the material jetting pattern. The output of the UV lamp and the rotational speed of the roller (600 rpm) were held constant during printing. This means that the material shore hardness is not changed by the output of the UV lamp and the speed of the roller.

3. There is no mention of elastic modulus of the material that was used to print the cornea and the holder. Furthermore, as the cornea is a viscoelastic material, a rubber-like material (hence a purely elastic material) would not mimic corneal mechanical behavior. 

We have added the information of elastic module in printed-corneal and -holder to Table 1. Unfortunately, the young’s modulus of elasticity of the corneal was not disclosed because of the confidential policy. Our strength is that wet lab style using the animal or human eyes is not required, thus our artificial eye balls using 3-D printing is useful for the evaluation of IOP in basic research. Since the difference between the corneal material and rubber-like material is a further limitation, we tried to mimic corneal behavior in terms of elastic modulus.

Table 1. Material parameters of the eye model.

 Cornea (500 μm) Vitreous Holder

Product name Stratasys FLX9740-DM Stratasys Gel Support Stratasys VeroClearTM

Stiffness of material Rubber-like Gel Hard

Tensile strength 3-4 MPa - 50-65 MPa

Young’s modulus of elasticity Un open (industrial secrets) - 2000-3000 Mpa

 Elongation break 190-210 % - 10-25 %

Tensile tear resistance 6.0-8.0 kg/cm - -

Flexural strength Un open (industrial secrets) - 75-110 Mpa

Flexural module Un open (industrial secrets) - 2200-3200 Mpa

4. The authors should have reported intraclass coefficient of serial measurements instead of describing repeated measurements as having "good repeatability".

Based on the calculation of intraclass correlation coefficients (ICCs), we showed the good repeatability in IOP measurements. The repeatability of IOP was high with 0.969 of an ICC (95% confidence interval: 0.909 to 0.994).

[Page 6, lines 101-105]: Two sentences have been added.

“To determine the repeatability of the IOP measurement, intraclass correlation coefficients (ICCs) were calculated for the 5 repeated measurements at each corneal thickness. The ICC ranges from 0 to 1 and is commonly classified as follows: less than 0.75 indicates poor repeatability, 0.75 to less than 0.90 shows moderate repeatability, and greater than 0.90 represents high repeatability.”

[Page 7, lines 119-121]: One sentence has been added.

“The repeatability of the IOP measurement was high, with an ICC of 0.969 (95 % confidence interval: 0.909 to 0.994).”

5. Although acknowledged as a limitation, failure to factor in lens, iris, aqueous humor is a significant limitation hindering the applicability of this approach in studying IOP dynamics ex vivo.

As you mentioned, our eye models did not have the components of iris, aqueous humor, and lens. These factors affect the IOP dynamics in human. It was hard to realize the precise 3D-printed eyeball model because of the technological limitation.

[Page 10, lines 167-169]: Two sentences have been added.

“Third, our 3D-printed eyeball models did not include iris, aqueous humor, or lens components, all of which affect IOP dynamics in humans, due to technological limitations.”

 

To Reviewer #2:

The authors present a new methodology to study the impact of corneal thickness on the intraocular pressure (IOP) measurements. Overall, the paper addresses a quite interesting problem; on the other hand, there are several issues which must be addressed prior to the paper acceptance.

Thank you for your positive comments on the revision. We have addressed the reply to your comments to enhance the quality of the paper.

At first, the introduction and general contextualization is quite poor. For example, the authors cite that numerous published studies have considered 3D printing techniques in Health Sciences; however, just a few were cited. Besides, it would be interesting to mention other studies which considered using 3D printing to bring digital models to real life and then perform tests on them. They may consider, for example:

- Combining Microtomography, 3D Printing, and Numerical Simulations to Study Scale Effects on the Permeability of Porous Media by Luan C. de S. M. Ozelim and André L. B. Cavalcante

- A 3D additive manufacturing approach for the validation of a numerical wall-scale model of catalytic particulate filters by Igor Belot, Yixun Sun, David Vidal, Martin Votsmeier, Philippe Causse, François Trochu, François Bertrand.

We have revised and added some sentences in INTRODUCTION. The two references you recommended have been added.

[Page 3, lines 42-49]: Three sentences have been added.

“Three-dimensional printing is an all-inclusive term for a variety of methods that use digital data to produce 3D objects made of various materials (both synthetic and organic) [2,3]. This technology can rapidly produce customized devices and prostheses at low costs via simple development and design processes and is capable of a single item at a time, each unique in shape and design. Such products may be as diverse as instruments that aid in early detection of common ocular conditions, diagnostic and therapeutic devices built specifically for individual patients, 3D-printed contact lenses and intraocular implants, and models that assist in surgery planning, improve patient and medical staff education.”

[Page 11, lines 181-186]: Two references have been added.

“2. Ozelim, Luan C. de SM, and André LB Cavalcante. "Combining microtomography, 3D printing, and numerical simulations to study scale effects on the permeability of porous media." International Journal of Geomechanics 19.2 (2019): 04018194.”

“3. Belot, Igor, et al. "A 3D additive manufacturing approach for the validation of a numerical wall-scale model of catalytic particulate filters." Chemical Engineering Journal 405 (2021): 126653.”

English level should be enhanced. Sometimes, it becomes a bit confusing to fully understand the sentences.

We have revised English in this revision by a native reviewer.

Regarding the 3D printed model, more information should be provided about the elastic properties of the materials used. For example, which are the Young moduli of the materials? This, together with the corneal thickness, would probably have an impact on the IOP measurements. Besides, it is important to fully compare the mechanical properties of the printing materials to the ones of real human eyes.

We have added the information of elastic module in printed-corneal and -holder to Table 1. Unfortunately, the young’s modulus of elasticity of the corneal was not disclosed because of the confidential policy. Our strength is that wet lab style using the animal or human eyes is not required, thus our artificial eye balls using 3-D printing is useful for the evaluation of IOP in basic research. Since the difference between the corneal material and rubber-like material is a further limitation, we tried to mimic corneal behavior in terms of elastic modulus.

Table 1. Material parameters of the eye model.

 Cornea (500 μm) Vitreous Holder

Product name Stratasys FLX9740-DM Stratasys Gel Support Stratasys VeroClearTM

Stiffness of material Rubber-like Gel Hard

Tensile strength 3-4 MPa - 50-65 MPa

Young’s modulus of elasticity Un open (industrial secrets) - 2000-3000 Mpa

 Elongation break 190-210 % - 10-25 %

Tensile tear resistance 6.0-8.0 kg/cm - -

Flexural strength Un open (industrial secrets) - 75-110 Mpa

Flexural module Un open (industrial secrets) - 2200-3200 Mpa

The authors should provide more information about the printing procedure. Which were the steps? Also, they should present more details about the printer and its specs (minimum detail, minimum wall thickness etc).

This method of additive manufacturing is “Material Jetting”.

This process is the following:

1. Jetting some liquid materials which are ultra violet (UV) curable resin on the print tray.

2. Making level UV curable resin by roller.

3. Hardening UV curable resin by UV lamp.

4. Repeat process 1-3 and building models.

Material shore hardness is changed by material jetting pattern. Output of UV lamp and 600 rpm of roller are constant during printing. It means Material shore hardness isn’t changed by output of UV lamp and rpm of roller. As for its specs (minimum detail, minimum wall thickness etc), Stratasys Japan Co. did not disclose the information because of a confidential policy.

[Page 4, lines 71-80]: Four sentences have been added.

When we manufactured the eyeball, the “Material Jetting” style was used as an additive manufacturing. This process is as follows:

1. Jet liquid ultra violet (UV) curable resin onto the print tray.

2. Use a roller to level the UV curable resin.

3. Use a UV lamp to harden the UV curable resin.

4. Repeat processes 1-3 and build the models.

The material shore hardness is changed by the material jetting pattern. The output of the UV lamp and the rotational speed of the roller (600 rpm) were held constant during printing. This means that the material shore hardness is not changed by the output of the UV lamp and the speed of the roller.

A key aspect which needs to be at least discussed is how the authors assessed the quality of the printed specimens. In short, what are the expected deviations between the printed specimen and the one in the CAD design? Will these deviations impact on the effective corneal thickness? I do not know if the authors have access to a CT scan, but if they do, scanning a few of the printed samples would bring a valuable validation to the thicknesses considered in their analyses.

In this revision, to assess the deviation between the printed specimen and the one in the CAD design, we tried to measure the actual value of CCT using an ultrasound pachymeter (SP-100; Tomey, Nagoya, Japan) (Table 3). Three consecutive measurements were performed in each eye model. The difference in CCT between the CAD and ultrasound style was clinically acceptable because of small difference (within 5 μm).

Table 3. Actual value of the central corneal thickness measured using ultrasound.

Central corneal thickness 

CAD design (μm) 200 300 400 500 600 700 800

Ultrasound pachymeter (μm) 201.3 ± 7.6 303.0 ± 6.2 403.7 ± 7.2 504.7 ± 7.4 601.7 ± 12.1 699.7 ± 11.5 803.0 ± 2.0

[Page 5, lines 87-89]: Two sentences have been added.

“To assess the deviation between the printed specimen and the CAD design, we aimed to measure the actual value of the CCT using an ultrasound pachymeter (SP-100; Tomey, Nagoya, Japan). Three consecutive measurements were performed in each eye model.”

[Page 8, line 127]: One sentence has been added.

“Table 3 presents the actual value of the CCT measured using an ultrasound pachymeter.”

[Page 9, lines 156-159]: Two sentences have been added.

“With regard to the reliability of the CCT in the CAD model, we evaluated the actual value of the CCT using an ultrasound pachymeter. The difference in the CCT between the CAD model and the ultrasound measurements was clinically acceptable because of the small difference (within 5 μm), as shown in Table 3.”

The authors mention the R-squared and the p-values obtained for the linear fit. Which hypothesis test is the p-value related to?

The normality of all data was first checked by the Kolmogorov-Smirnov test. Because the use of parametric statistics was possible, the Pearson correlation coefficient was used to assess the correlation of the IOP with CCT in each eye model.

[Page 7, lines 110-113]: Two sentences have been added.

“The normality of all data was first checked by the Kolmogorov‒Smirnov test. Because the use of parametric statistics was possible, the Pearson correlation coefficient was used to assess the correlation between the IOP and the CCT in each eye model.”

Since the authors have a complete control over the model, are they capable of measuring the “true” exact IOP?

Exact answer to the true IOP might be unknow because it is impossible to determine the exact IOP which represents the pressure at optic disc. Generally, corneal biomechanics plays a role in IOP in human eye. Thus, we mimicked the different IOP model with increasing CCT using 3D-printed eye models.

[Page 8, lines 136-140]: Two sentences have been added.

“However, the true IOP might be unknown because it is impossible to determine the exact IOP that represents the pressure at the optic disc. Generally, corneal biomechanics play a role in the IOP in the human eye. Thus, we mimicked different IOP models with increasing CCTs using 3D-printed eye models.”

Since Tono-Pen AVIA has not been calibrated and designed to be used with the 3D printed materials, this would be interesting to actually confirm the measurements are correct. Indicating that the results are reproducible is not sufficient to indicate the correctness of the result itself.

As you mentioned, it is hard for Tono-Pen AVIA to determine the correctness of IOP measurements in 3D-printed eye models because Tono-Pen AVIA itself has not been calibrated and designed to be used with those models. We should have compared the IOP with other tannometers such as iCare and non-contact tonometer.

[Page 9, lines 152-156]: Two sentences have been added.

“However, it is difficult for the Tono-Pen AVIA to determine the correctness of the IOP measurements in 3D-printed eye models because the Tono-Pen AVIA itself has not been calibrated and designed to be used with those models. It would have been more beneficial to compare the IOP measured with other tannometers, such as iCare and noncontact tonometers.”

Finally, do the authors have control over the IOP change? How? Are the changes observed only due to the CCT changes? Any additional pressure is later applied to the gels? It is not clear in the paper how the IOP change occurs.

To simply control the change in the IOP, we changed the only CCT in the current study. The other components of model eyes were identical to each model. 

[Page 5, lines 85-86]: One sentence has been added.

“To simply control the change in the IOP, we changed only the CCT in the current study.”

The issues above must be addressed to enhance the quality of the paper and make it publishable in such a high standard journal as PLOS One.

6. PLOS authors have the option to publish the peer review history of their article (what does this mean?). If published, this will include your full peer review and any attached files.

Do you want your identity to be public for this peer review? For information about this choice, including consent withdrawal, please see our Privacy Policy.

Reviewer #1: Yes: Eray Atalay

Reviewer #2: No

We believe the manuscript has been prepared and submitted satisfactorily and hope that it will be accepted for publication in Plos One. Thank you for your attention and consideration.

Sincerely yours,

Hidenaga Kobashi, MD, PhD, Department of Ophthalmology, Keio University, School of Medicine, Tokyo, Japan. E-mail address: hidenaga_kobashi@keio.jp

---

## [Decision Letter · Decision Letter 1]

27 Feb 2023

3D-printed eye model: simulation of intraocular pressure

PONE-D-22-23613R1

Dear Sir,

We’re pleased to inform you that your manuscript has been judged scientifically suitable for publication and will be formally accepted for publication once it meets all outstanding technical requirements.

Kind regards,

Aparna Rao

Academic Editor

PLOS ONE

Reviewers' comments:

Reviewer's Responses to Questions

**Comments to the Author**

1. If the authors have adequately addressed your comments raised in a previous round of review and you feel that this manuscript is now acceptable for publication, you may indicate that here to bypass the “Comments to the Author” section, enter your conflict of interest statement in the “Confidential to Editor” section, and submit your "Accept" recommendation.

Reviewer #2: All comments have been addressed

2. Is the manuscript technically sound, and do the data support the conclusions?

Reviewer #2: Yes

3. Has the statistical analysis been performed appropriately and rigorously? 

Reviewer #2: Yes

4. Have the authors made all data underlying the findings in their manuscript fully available?

Reviewer #2: Yes

5. Is the manuscript presented in an intelligible fashion and written in standard English?

Reviewer #2: Yes

6. Review Comments to the Author

Reviewer #2: All the comments have been addressed in the new version of the manuscript. Therefore, I advise to accept the paper in its current form.

7. PLOS authors have the option to publish the peer review history of their article (what does this mean?). If published, this will include your full peer review and any attached files.

Reviewer #2: No

---

## [Editor Report · Acceptance letter]

1 Mar 2023

PONE-D-22-23613R1 

3D-printed eye model: simulation of intraocular pressure 

Dear Dr. Kobashi:

I'm pleased to inform you that your manuscript has been deemed suitable for publication in PLOS ONE. Congratulations! Your manuscript is now with our production department. 

Kind regards, 

on behalf of

Dr. Aparna Rao 

Academic Editor

PLOS ONE